# Targeted modulation of protein liquid–liquid phase separation by evolution of amino-acid sequence

Simon M. Lichtinger[1¤], Adiran Garaizar[2], Rosana Collepardo-Guevara[1,2,3]*, Aleks Reinhardt[1]*

1 Yusuf Hamied Department of Chemistry, University of Cambridge, Cambridge, United Kingdom, 2 Department of Physics, Cavendish Laboratory, Maxwell Centre, University of Cambridge, Cambridge, United Kingdom, 3 Department of Genetics, University of Cambridge, Cambridge, United Kingdom

¤ Current address: Department of Biochemistry, University of Oxford, Oxford, United Kingdom
* rc597@cam.ac.uk (RC-G); ar732@cam.ac.uk (AR)

**Data Availability Statement:** All relevant data are within the manuscript and its Supporting information files.

## Abstract

Rationally and efficiently modifying the amino-acid sequence of proteins to control their ability to undergo liquid–liquid phase separation (LLPS) on demand is not only highly desirable, but can also help to elucidate which protein features are important for LLPS. Here, we propose a computational method that couples a genetic algorithm to a sequence-dependent coarse-grained protein model to evolve the amino-acid sequences of phase-separating intrinsically disordered protein regions (IDRs), and purposely enhance or inhibit their capacity to phase-separate. We validate the predicted critical solution temperatures of the mutated sequences with ABSINTH, a more accurate all-atom model. We apply the algorithm to the phase-separating IDRs of three naturally occurring proteins, namely FUS, hnRNPA1 and LAF1, as proto-types of regions that exist in cells and undergo homotypic LLPS driven by different types of intermolecular interaction, and we find that the evolution of amino-acid sequences towards enhanced LLPS is driven in these three cases, among other factors, by an increase in the average size of the amino acids. However, the direction of change in the molecular driving forces that enhance LLPS (such as hydrophobicity, aromaticity and charge) depends on the initial amino-acid sequence. Finally, we show that the evolution of amino-acid sequences to modulate LLPS is strongly coupled to the make-up of the medium (e.g. the presence or absence of RNA), which may have significant implications for our understanding of phase separation within the many-component mixtures of biological systems.

## Author summary

Protein condensates formed by the process of liquid–liquid phase separation (LLPS) play diverse roles inside cells—from spatio-temporal compartmentalisation to speeding up chemical reactions. When things go wrong, LLPS can have pathological implications. This realisation has boosted the interest in devising approaches to design rationally amino-acid sequence variations to modulate or even reverse the phase behaviour of proteins on

**Funding:** RCG received funding from Science and Technology Facilities Council (DiRAC), dirac.ac.uk, under grant 803326 from European Research Council (Horizon 2020), erc.europa.eu and from Winton Programme for the Physics of Sustainability, www.winton.phy.cam.ac.uk AG received funding under grant EP/N509620/1 from Engineering and Physical Sciences Research Council, epsrc.ukri.org AR and RCG received funding under grant EP/P020259/1 [Tier-2 projects cs061, cs104] from Engineering and Physical Sciences Research Council, epsrc.ukri.org SML received funding from German Academic Scholarship Foundation, www.studienstiftung.de The funders had no role in study design, data collection and analysis, decision to publish, or preparation of the manuscript.

**Competing interests:** The authors have declared that no competing interests exist.

demand. Here, we develop an efficient computational method that combines a genetic algorithm with a sequence-dependent coarse-grained model, and an all-atom model for validation, to identify amino-acid sequence variations of intrinsically disordered proteins that intentionally promote or inhibit their LLPS. Our method can be applied to proteins in pure form and within multi-component systems.

This is a *PLOS Computational Biology* Methods paper.

## Introduction

Liquid–liquid phase separation (LLPS) of multivalent biomolecules (e.g. proteins and nucleic acids) is an important mechanism employed by cells to control the spatio-temporal organisation of their many components [1, 2]. Biomolecular condensates, or membraneless organelles, such as stress granules [3], P-granules [4, 5], the nephrin–NCK–WASP system [6] and the nucleoli [7], are formed by LLPS and have diverse biological functions. LLPS inside cells plays a very diverse range of roles beyond membraneless compartmentalisation, such as in gene silencing via heterochromatin formation [8–10], in gene activation by facilitating the formation of super-enhancers [11], in buffering cellular noise [12], in modulating enzymatic reactions [13] and in sensing pH changes in the skin [14]. However, some biomolecular condensates emerge spontaneously inside cells without as-yet clearly identified functions; it has been hypothesised that some of these might be implicated in the emergence of phase-separation-related pathologies [15]. Indeed, aberrant LLPS of the proteins Fused in Sarcoma (FUS) and Tau has been associated with the onset of degenerative diseases such as amyotrophic lateral sclerosis and Alzheimer's disease, respectively [15]. More recently, biomolecular condensates have been proposed as promising new tools to partition anti-cancer drugs preferentially to cancer cells [16]. Such a richness of behaviours highlights the importance of learning to design protein mutations that can alter the stabilities of condensates.

When designing protein mutations, it is useful to consider that the thermodynamics of phase separation is driven by the competition between interaction enthalpies and the entropic favourability of mixing [17–19]. In the context of protein solutions at physiological conditions, LLPS is principally stabilised by π-stacking and cation–π interactions, followed by charge–charge, dipole–dipole and other hydrophobic interactions [20–23]. The relative contributions of different amino-acid pair interactions to LLPS stability is further modulated by the experimental conditions, including the temperature, pH and salt concentration [21]. In biomolecular systems, the multivalency in mixtures is thus the main physical parameter that defines the ability of a system to undergo LLPS [6, 15, 22, 24–26]: biomolecules with higher valencies can establish a larger number of weak attractive interactions with other species and hence form a more stable condensate.

The connections between LLPS and cellular function, and between aberrant LLPS and human pathologies, suggest that learning how to control or even prevent the phase separation of proteins by subtly mutating their amino-acid sequences would be highly desirable. However, the sequence space of even the smallest naturally occurring proteins is immense, which makes the task of choosing mutations manually extremely inefficient; what is more, even if small-scale modifications of a single protein that promote phase separation might be possible to design manually with some physical intuition, biomolecular LLPS is a collective phenomenon involving many weak interactions, and it is not at all straightforward to anticipate how small sequence modifications affect the phase behaviour of a protein mixture without the use of an algorithm.

Indeed, optimising biological LLPS is especially difficult because *in vivo* biomolecular condensates can be highly multi-component systems [25, 27–29]. Furthermore, over 270 distinct multivalent proteins have been shown to undergo LLPS *in vitro* [30]. Despite this complexity, the properties of condensates can be successfully approximated in vitro by considering just a fraction of biomolecules, known as 'scaffolds', since such molecules tend to dominate the phase behaviour [6, 24, 31]: the addition of biomolecules that are recruited to condensates via their interactions with scaffolds, termed 'clients' [6, 24], impacts the stability of condensates only marginally [26], making the problem somewhat more manageable.

A wide body of work has significantly advanced our understanding of how changes in amino-acid sequence transform the phase behaviour of different proteins [22, 23, 32–38]. Notably, tightly integrated experiments and simulations with the 'stickers and spacers' model explain how changing the number and patterning of aromatic and charged residues can alter the phase diagrams of prion-like-domain proteins in a predictable manner [22, 23]. Experiments also demonstrate that multimerising the arginine/glycine-rich RGG domain of LAF1 leads to controllable phase separation [39], and minimal coarse-grained simulations of point mutations of two designer proteins exemplify how on-demand modulation of protein phase behaviour can be achieved *in vivo* [40].

Alongside globular domains, intrinsically disordered regions (IDRs) are thought to be one of the main drivers of LLPS in protein systems [41, 42]. Various theoretical approaches, including random-phase approximation theory, have been applied to LLPS of IDRs [43, 44]. Such treatments can rationalise aspects of charge distribution in phase-separating IDRs, and their relative simplicity has allowed us to gain significant intuition for the electrostatic aspects of phase separation. However, the scope of theory is limited by the inherent approximations that are of necessity made and the sheer complexity of biological multi-component systems. More realistic representations of proteins are usually too complex for analytical treatment, but can be studied in computer simulation, giving us molecular-level insight into the phase behaviour of biological systems. Computational approaches hold significant promise of enabling us to probe amino-acid sequence space efficiently and to design protein mutations which enhance or inhibit LLPS of a protein solution. Although all-atom simulations of LLPS in explicit solvents are only now slowly becoming computationally tractable [45–48] due to the exponential scaling of search space with system size, the ABSINTH ('self-Assembly of Biomolecules Studied by an Implicit, Novel, Tunable Hamiltonian') all-atom and implicit solvent framework [49, 50] of Pappu and colleagues has been consistently applied to estimate relative critical solution temperatures from single-molecule properties and to capture subtle variations in sequence space in agreement with experiment [22, 51]. Most recently, ABSINTH has been successfully paired with Gaussian cluster theory to compute full sequence-dependent phase diagrams of proteins [51].

Theory and simulations using coarse-grained potentials—from patchy particles to lattice models [22, 26, 35, 39, 45, 49, 52–58]—have provided significant microscopic insight into the physics of phase separation. Amongst the growing body of coarse-grained models available to investigate protein phase behaviour computationally, a noteworthy approach is the 'stickers-and-spacers' model of Pappu and colleagues. The stickers-and-spacers model represents multivalent proteins as heteropolymers composed of stickers (LLPS-binding motifs) and spacers (regions in between stickers), and has been used to generate phase diagrams of proteins in perfect agreement with experiment and to elucidate the underlying molecular driving forces [22, 36, 55]. The residue resolution HPS ('HydroPhobicity Scale') and KH ('Kim–Hummer') coarse-grained models of the Mittal and Best groups [35, 59], combined with the direct-coexistence simulation method, also stand out among the techniques to assess the effect of amino-acid sequence variation on protein LLPS. Some of the advantages of the HPS/KH models

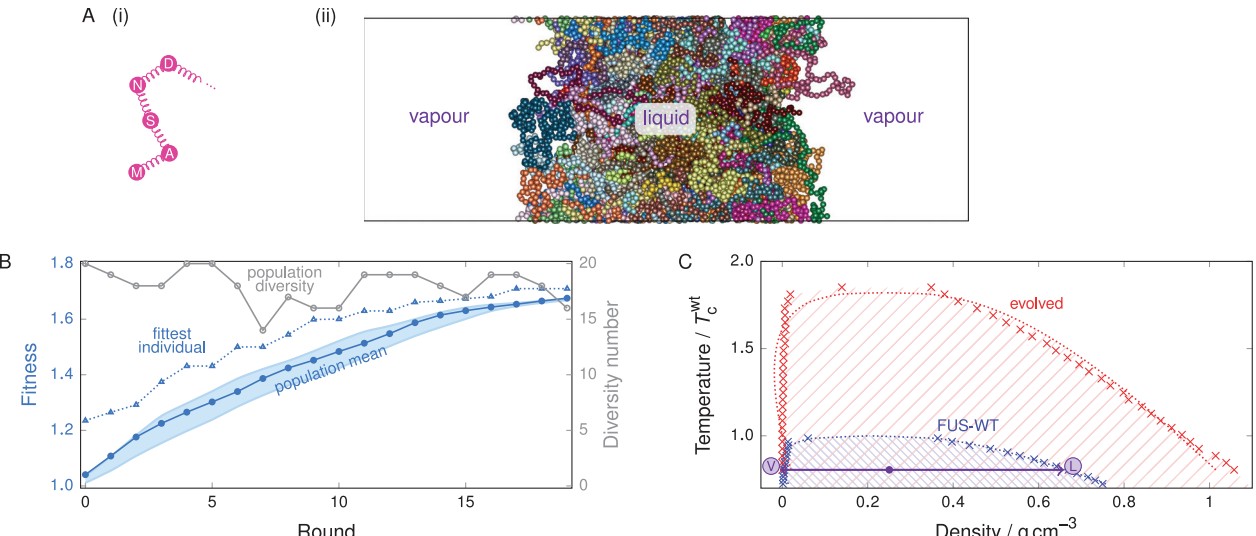

**Fig 1. Evolution behaviour of FUS PLD. (A)** (i) A schematic representation of the model used. Each amino acid is represented by a bead, and beads are connected with harmonic springs. (ii) A snapshot of a typical simulation cell exhibiting coexistence between a liquid-like (protein-rich, high-density) fluid and a vapour-like (protein-poor, low-density) fluid. The box is periodic in all directions. Different colours are used to represent beads in different protein chains. **(B)** Typical genetic-algorithm progression for FUS where the fitness function *increases* the width of the phase diagram. The fitness function [Eq (2)] increases by ∼65% over 20 rounds. The fittest individual is 5% to 20% fitter than the mean in most rounds. The population diversity, i.e. the number of distinct sequences present in the overall population of 20, remains high throughout the run. The shaded area corresponds to the range of values of the mean fitness obtained from 3 independent genetic-algorithm runs. **(C)** Comparison of representative phase diagrams before and after genetic-algorithm runs, confirming that the fitness function choice was suitable. Pale hatched lines indicate the approximate region of phase separation for each case. Error bars in the density evaluations are smaller than the symbols, and dotted lines are fits as detailed in S1 Text (Sec. S1-b). The point labelled in violet corresponds to the snapshot shown in panel **(A)**, with the densities of the vapour-like and liquid-like fluids labelled 'V' and 'L', respectively. All densities reported are for the protein in a massless implicit solvent.

include their transferability and their ability to compute phase diagrams of large proteins (i.e. up to ∼500 residues per protein) at residue resolution.

Here, we develop a genetic-algorithm approach coupled to the sequence-dependent coarse-grained model of proteins with amino-acid resolution of the Mittal and Best groups [35] [Fig 1A(i)] to design protein mutations that can enhance or inhibit LLPS, and use the all-atom implicit solvent ABSINTH framework [49, 50] to verify the validity of our predictions. Our method takes advantage of the computational efficiency of the residue-resolution coarse-grained models to search sequence space, and the higher accuracy of ABSINTH to predict experimentally-consistent critical temperature of proteins. Genetic algorithms have been used since the early 1990s with considerable success in a variety of fields, from reaction dynamics [60, 61] to crystal and cluster structure prediction [62, 63], protein evolution [64–66] and drug design [67], including when coupled with computer simulations [68–73]. In the past year, the integration of genetic algorithms and coarse-grained models applied to biological questions seems to be gaining traction [73, 74]; indeed, very recently, a genetic-algorithm approach has been used to design sequences of proteins that exhibit lower critical solution temperatures [74]. In our implementation, we anchor a genetic algorithm to a fitness function that is fast enough to be evolved and that represents a good proxy for the critical solution temperature, which measures the ability of a protein to phase-separate. With this approach, we systematically evolve the amino-acid sequences of the IDRs of three naturally occurring proteins that are known to phase-separate *in vitro* via homotypic interactions, and we show that we can drive the genetic algorithm either to enhance or to inhibit their LLPS. By shuffling the amino-acid sequences in chunks of varying lengths, we also identify the binding domains of the IDRs

that are essential to drive LLPS (the 'stickers') and the connecting regions (the 'spacers') [22, 36, 55]. By investigating LLPS in the vicinity of known phase-separating sequences, we can infer which features of a sequence drive phase separation in biological systems. While previously, artificial sequences have been probed in a systematic way, for instance in the context of charge patterning [45], our work also complements very recent results obtained on LAF1-IDR [39] and Ddx4-IDR [58]. Although some of the fine features of our findings may be model-specific, we validate the robustness of our genetic algorithm by repeating core runs using an alternative parameterisation of the protein coarse-grained model [58]. We also benchmark the model's capacity to predict critical solution temperatures against experimental data [23] and the ABSINTH implicit solvation model and all-atom force-field paradigm [49, 50].

## Results

### Protein phase behaviour can be guided by a genetic algorithm

A free choice from amongst the 20 canonical amino acids in a protein with $n$ residues amounts to an $n$-dimensional vector with $20^n$ possible sequences, where for each sequence one might attempt to compute some property that characterises the sequence's LLPS behaviour. One possible quantity that can serve this purpose is the upper critical solution temperature $T_c$, above which no de-mixing occurs. However, an exhaustive search of sequence space would be prohibitively expensive. Moreover, finding the 'optimal' critical temperature, however we might choose to define it, is not itself the aim. For example, a poly-F chain has a particularly high $T_c$ (see S1 Text (Sec. S2)) in the protein model we have used, but studying it in detail is not particularly helpful in understanding what drives biological phase separation. We instead focus on biologically occurring proteins and evolve their amino-acid sequences with the aim of finding individual examples of sequences that either extend or narrow (as opposed to maximise or minimise) the range of thermodynamic conditions where homotypic LLPS occurs; in other words, our goal is to use the genetic algorithm to perform only local optimisation. In particular, we are interested in the effect of relatively small changes to the amino-acid sequence on phase separation, as these are instrumental to understanding how modifications can be designed to control the phase behaviour of proteins *in vivo*. Moreover, such modified sequences might more easily be introduced into cells.

To determine sets of mutations that shift the phase behaviour of a protein in the desired direction, we start from a reference amino-acid sequence and perform direct-coexistence molecular dynamics (MD) simulations of a sufficiently large number of copies of that protein (see Fig 1A(ii) and S1 Text (Sec. S1-b)). In direct-coexistence simulations, we explicitly simulate two different phases—a protein-enriched solution and a protein-depleted solution—in contact with each other in the same simulation box. By performing such simulations at several temperatures, we can approximate the compositional phase diagram, which indicates which phase is thermodynamically stable as a function of temperature and protein density. We then seek to evolve the protein towards enhanced or inhibited LLPS by developing a genetic algorithm (see Methods) that iteratively proposes stochastic amino-acid sequence mutations, selecting a few at each iteration amongst those that induce the strongest effect amongst the set in the protein LLPS, and mutating again. For a given protein model, whether such a genetic-algorithm approach can succeed in evolving protein phase behaviour depends on the quality and efficiency of the fitness function used to control it. Our genetic algorithm uses the difference in composition densities of the protein-poor and protein-rich phases at constant volume as its fitness function. As we show below, our fitness function is both computationally inexpensive and a good metric to determine whether a set of mutations would result in enhanced or inhibited LLPS. Although the critical solution temperature of a protein mixture may seem like

an obvious order parameter to determine whether a specific set of mutations promotes or inhibits LLPS—by raising and lowering the critical solution temperature, respectively—computing it in every round of the evolution process when using a residue-resolution coarse-grained protein model is computationally infeasible. This is because estimating the critical solution temperature requires an evaluation of a full phase diagram of the protein solution, which in turn requires either the use of very expensive free-energy methods [75], or performing direct-coexistence simulations at a number of different temperatures, each involving long MD simulations of a large number of copies of the same protein, and analysing the results to extrapolate the data and estimate the critical temperature. By contrast, evaluating the difference in composition densities requires only one set of direct-coexistence simulations to be run at a fixed sub-critical temperature (i.e. below $T_c$).

## The case of the PLD of FUS

**The range of stability of LLPS can be evolved.**   As an initial model system, we investigate the behaviour of the prion-like domain (PLD) of the FUS protein, an IDR rich in tyrosines and mostly devoid of charged residues. Although the PLD of FUS only phase-separates *in vitro* at somewhat extreme conditions with respect to the physiological ones (namely low salt concentrations of 37.5 mM NaCl and high protein concentrations of 6 μM to 33 μM) [76], PLD–PLD interactions and PLD–arginine-rich domain interactions drive LLPS of the full FUS protein under physiological conditions, both *in vitro* and in cells [77]. We first use direct-coexistence molecular dynamics simulations to approximate the compositional phase diagram (i.e. in the temperature versus protein-density space) of the PLD of FUS. We then seek to evolve the system's critical solution temperature by introducing our genetic algorithm (see Methods), which allows us to mimic, broadly speaking, the evolutionary pathways that might drive phase separation in nature. Starting from the reference amino-acid sequence of FUS PLD ('WT-FUS') given in S1 Text (Sec. S6), we use our genetic algorithm to attempt separately to increase and to reduce the width of the binodal curve of the compositional phase diagram. We show the population fitness [Eq (2)] and the diversity of the population as functions of the genetic-algorithm round in Fig 1B for the case of increasing the phase diagram width; analogous results for the case of reducing it are shown in S4 Fig. The genetic algorithm is effective in increasing the population fitness in each case, and in both cases the population diversity remains high, indicating that no premature convergence occurs. We also confirm the effectiveness of the driven evolution in our genetic algorithm by contrasting the results to a dummy variant with all the mutagenesis steps intact, but the selection pressure abolished [S1 Text (Sec. S3)]. Furthermore, over the three repeats of the genetic-algorithm runs we performed to increase the phase diagram width, mutation of all but two residues (27 and 155) was attempted by the genetic algorithm at least once, which indicates a sufficient mutation rate to probe the entire sequence over 20 rounds. In Fig 1C, we show that the phase diagram of the evolved FUS PLD in the former case exhibits a large increase in the range of temperatures and densities at which LLPS occurs, with the critical temperature increasing by ∼65% compared to the WT-FUS PLD. Although we have used the width of the phase diagram as a proxy for the critical solution temperature, Fig 1C confirms not only that the critical solution temperature behaves in the expected way, but also that genetic algorithms with simple fitness functions can significantly perturb the LLPS behaviour, leading to an effective gradient in sequence space. These results suggest that a genetic algorithm can be used to search the sequence space of proteins efficiently and can help propose sequence mutations that yield meaningful changes in the proteins' compositional phase diagrams. Importantly, as mentioned earlier, our approach is computationally tractable because our goal is to find candidate mutations that can purposely increase or narrow

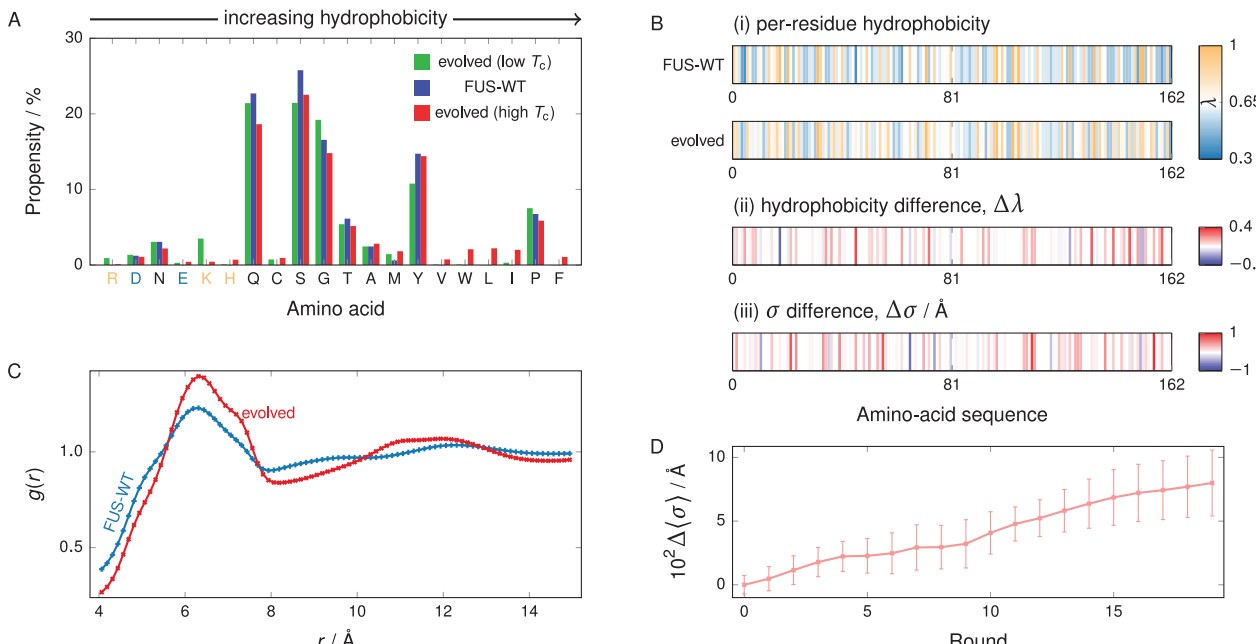

**Fig 2. Properties of evolved FUS PLD. (A)** Amino-acid composition before and after applying the genetic algorithm to increase and reduce the phase diagram width, and hence the critical temperature, starting from WT-FUS. Amino acids are plotted in order of increasing hydrophobicity [see Table A in S1 Text]. Positively charged amino acids are indicated in light orange and negatively ones in light blue. In genetic-algorithm runs which increase $T_c$, hydrophobic amino acids become favoured, whilst the converse holds for genetic-algorithm runs which decrease $T_c$. **(B)** Map of (i) the per-residue hydrophobicity along the sequence, (ii) the change from the wild type to the evolved protein after evolution towards higher $T_c$ and (iii) the change of the per-residue amino-acid size ($\sigma$). The data for the evolved protein are averaged over the entire population at the end of 3 independent genetic-algorithm runs. No larger-scale regional preference for modification is readily apparent. Trends in hydrophobicity and $\sigma$ are largely correlated. **(C)** Comparison of the pair correlation function $g(r)$ before and after evolving FUS with a genetic algorithm towards a larger phase diagram width. The data were computed at the same bead number density $N/V = 7.0$ nm$^{-3}$ and temperature $T = 0.8 T_c^{\mathrm{wt}}$. In both cases, this temperature is below the critical point and the density corresponds to the protein-rich (liquid-like) phase, i.e. a point that lies above the binodal line on the phase diagram. We compute $g(r)$ by finding for each bead $i$ in the system the number of all non-harmonically bonded other beads within a distance $r + \delta r$ of the each other for bins of width $\delta r$, averaging over each bead $i$ in the system, and normalising the result by the volume element and the (common) number density. The symbol size is larger than the standard deviation of the average across 4 independent simulations. In the case of the evolved system, the more pronounced nearest-neighbour maximum indicates the local environment is more structured than in the case of WT-FUS. **(D)** The average $\sigma$ value of the amino acids in the population increases over the course of the genetic-algorithm run. Error bars are standard deviations of the averaged $\sigma$ value of individual sequences with respect to the pooled population from the three genetic-algorithm runs.

the range of stability of LLPS, rather than to identify the specific amino-acid sequences that give rise to a true maximum or minimum critical solution temperatures, and hence, performing just a few iterations of the algorithm is sufficient.

**LLPS evolution can be driven by changes in hydrophobic and aromatic residue composition.** We analyse the extent to which all interactions other than direct charge–charge interactions, which here we term collectively as 'hydrophobicity', govern the evolution of the phase behaviour of proteins by estimating each amino acid's relative degree of hydrophobicity. We use the hydrophobicity scale proposed in [78], which is quantified as the $\lambda$ parameter in the coarse-grained model of [35], and which can be used to scale the well depth of the modified Lennard-Jones potential in an amino-acid-specific way (see S1 Text (Sec. S1-a)). In Fig 2A, we show the amino-acid compositions of the populations resulting from genetic-algorithm runs in which $T_c$ is increased and runs in which it is decreased, broken down by amino acid and ordered by the extent of hydrophobicity, alongside the reference WT sequence. The amino-acid sequences of the WT of the FUS PLD and examples of its evolved analogues are given in S1 Text (Sec. S7). In the case of runs targeting an increase in $T_c$, there is a general shift towards higher hydrophobicity, whilst the case where $T_c$ is targeted to decrease shows a trend towards

highly polar and charged amino acids. These trends in amino-acid composition confirm that, even though there are more strongly hydrophobic than weakly hydrophobic amino acids available for insertion, evolution of the FUS PLD is able to be driven in both the hydrophobic and hydrophilic directions. [One specific limitation of using a coarse-grained potential on results from a genetic-algorithm framework is its broadening effect on amino-acid composition: physically more important amino acids can stochastically be replaced by less important ones simply because their force-field parameters are similar. As a result, when we describe the strength of 'hydrophobic' interactions, we refer to the interactions in general between non-charged amino acids. However, a more accurate model to describe residue–residue interactions could in the future be coupled to our genetic algorithm to resolve such interactions in more detail.]

In addition to the attractiveness of the hydrophobic interactions, a further factor determining the strength of hydrophobic interactions is the size of each amino acid. We quantify this by $\sigma$, the Van der Waals radius of each amino acid (see Table A in S1 Text). In Fig 2D, we show that the average size of the amino acids in the sequence population increases as a function of the genetic-algorithm round, implying that the average size of the amino acids increases through evolution. Furthermore, as shown in Fig 2B(iii), although the effects on hydrophobic attractiveness and $\sigma$ values largely correlate at most residues of the protein sequence (with a Pearson coefficient of 0.42), this is not invariably the case, giving us the first indication that the size of the amino acids could play an independent role in determining LLPS properties. Although the overall increase in amino-acid size might be explained at least in part by the amino-acid size defining the range of the hydrophobic attractions, we hypothesise that the main physical driving force explaining the increase of both hydrophobicity and size is the ability of larger and more hydrophobic amino acids to form a more densely connected, and in turn more stable, condensed liquid-like protein-rich phase [26]. To test this hypothesis, we compute the pair correlation function of the protein-rich phase for both FUS-WT and one of the evolved sequences at a common number density [Fig 2C]. The nearest-neighbour maximum is 13% higher than in the wild type, indicating a greater degree of local structure and an increase in the number of nearest-neighbour beads compared to the WT, which has previously been shown [26] to correspond to a greater protein valency.

Because the sequence of the WT PLD of FUS only contains two (negatively) charged amino acids, its LLPS must be driven by hydrophobicity. However, our results show that the FUS sequence lies on a hydrophobic gradient in sequence space: an increase in hydrophobicity effects an increase in the critical solution temperature. It has been proposed [79, 80] that the driving force for the LLPS of this FUS IDR is specifically the interactions between tyrosine residues dispersed through the sequence. Although such interactions are only implicitly captured in the coarse-grained protein model we use through its hydrophobicity parameter, and thus the distribution of amino acids obtained follows broad trends rather than converging to a distinct amino acid or motif, our results are consistent with previous work [79, 81, 82], and meaningful trends in composition and sequence can be observed from our simulations. Our results thus suggest that evolving a protein sequence which is dominated by hydrophobic residues, as is the case for the PLD of FUS, towards enhancing its propensity for LLPS is efficiently achieved by protein mutations that increase the average attractiveness and size of the protein's uncharged residues.

**Hydrophobic patterning has a minimum length scale.**    In Fig 2B, we show how the accumulated changes in sequence, represented as the hydrophobicity of a residue, map onto the WT-FUS sequence. There are no larger-scale regions along the sequence where modifications occur preferentially; instead, there appears to be a stochastic increase in hydrophobicity, with less hydrophobic residues being replaced by more hydrophobic ones. However, since short

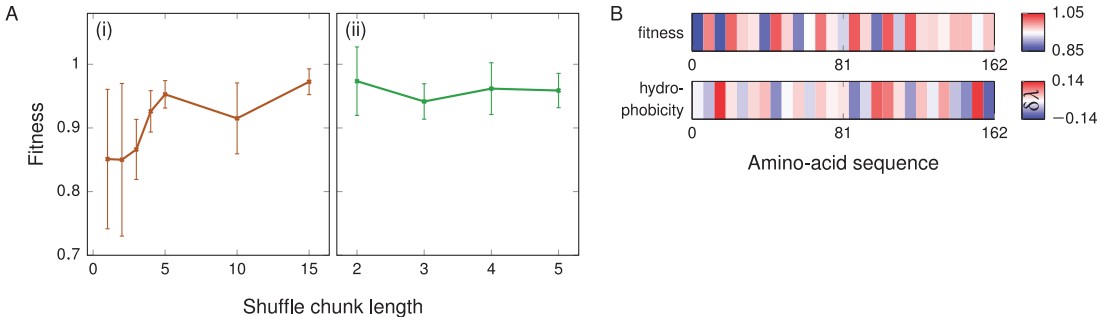

**Fig 3. Determining the characteristic length scale. (A)** (i) The fitness of the system as a function of the chunk length following the shuffling of approximately 100 amino acids. Error bars show the standard deviation across several shuffling runs. There is a significant difference in the fitness and the error bars for small chunks of 1 and 2 amino acids, whereas the error bars for the larger chunk sizes are considerably smaller. (ii) Analogous results for shuffling runs with a hydrophobicity bias, where exchanges were allowed only amongst the top 30% of chunks by hydrophobicity. Representative fitness functions as a function of the number of amino-acid pairs shuffled are shown in S5 Fig. **(B)** The value of the fitness function (relative to the WT fitness) when chunks of 6 amino acids of FUS-WT are separately replaced with glycine, and the average hydrophobicity of these same 6-residue chunks of FUS-WT relative to the hydrophobicity parameter of glycine, $\delta\lambda = \langle\lambda\rangle_{\text{chunk}} - \lambda(G)$, where $\lambda(G) = 0.649$. Where glycine represents a gain in hydrophobicity, the fitness change is largely positive, and vice versa.

runs of the genetic algorithm cannot result in perfectly uniform replacement attempt probabilities, we cannot expect to be able to resolve small-scale features in amino-acid sequence space. In order to investigate such small-scale features, we therefore first 'shuffle' the sequence without changing its overall amino-acid make-up. We choose chunks of varying lengths by randomly choosing two positions along the sequence and exchanging chunks of *l* amino acids starting from those two positions. The ends of the sequence are treated periodically to ensure no positional selection bias against the ends. The swapped amino acids may be of the same type. We record the fitness function of the protein sequences as a function of the total number of amino-acid pairs changed, i.e. the number of exchange steps multiplied by *l*. In Fig 3A, we show the variation of the fitness with chunk length after ∼100 amino acids have been shuffled. The error bar, which shows the standard deviation across several independent runs, is a useful measure of the sensitivity of the fitness function to amino-acid sequence. Very small chunk lengths, particularly of 1 or 2 amino acids, are highly disruptive to phase separation, while larger chunk lengths only cause smaller modulations. From these results, we can conclude that segments of 2–3 successive amino acids are crucial in driving LLPS in the PLD of FUS, representing the length scale of some sequence feature. To investigate the nature of this feature, we repeated the shuffling analysis with a hydrophobicity bias, where only the most hydrophobic of all possible contiguous chunks are exchanged. The dependence on chunk size largely disappears, implying that it was small hydrophobic patches that were previously disrupted by shuffling [Fig 3A(ii)]. The phase separation therefore appears to be governed by a hydrophobic patterning of a minimum length scale of 2–3 amino acids. This is consistent with the 'stickers-and-spacers' paradigm of phase-separating proteins, in which proteins are considered to comprise stickers—corresponding to the attractive protein regions that drive LLPS, in our case the small chunks of 2–3 amino-acid residues—that are connected by less attractive regions termed spacers [22, 36, 55].

Once we know this minimum length scale, we can investigate the effect of hydrophobicity by replacing successive chunks of the amino-acid sequence with a fixed amino acid. The conventional approach to probe the function of specific residues is alanine scanning [83–85]. As we are interested in how hydrophobicity affects phase behaviour, in this case, we mutate amino acids to glycine rather than alanine, as the former has the median hydrophobicity in the

coarse-grained protein model (see Table A in S1 Text). Although in experiments or all-atom simulations, such a replacement may be less appropriate, as glycine disrupts protein secondary structure by its dihedral angle preferences [86], in the CG model this effect is immaterial, as no conformational terms are considered. We replace successive chunks of 6 amino acids in each case; this chunk size is about 2–3 times the characteristic length scale of hydrophobic patterning, ensuring that differences observed most likely arise from the overall difference in hydrophobicity rather than a disruption of localised 'stickers'. Fig 3B shows the results of a glycine scan projected onto the chunk-averaged hydrophobicities of the WT-FUS sequence. The curves anti-correlate for most of the sequence (with a Pearson coefficient of −0.5), reflecting that in hydrophobic stretches, mutating to glycine decreases hydrophobicity and thus decreases fitness, while the converse holds for hydrophilic stretches, confirming the dominance of hydrophobicity as a driving force for LLPS in this case.

## Charge patterning may be an alternative driving force for evolution of protein phase behaviour

Not all proteins that exhibit LLPS are expected to be governed by the same driving force. For example, the patterning of charges has been suggested to contribute to LLPS in charge-rich proteins [44, 45], while the phase separation of the intrinsically disordered region of the protein hnRNPA1, which belongs to the family of heterogeneous nuclear ribonucleoproteins, has been shown to be driven by the interaction between linearly dispersed aromatic residues within the polar sequence [22, 23, 34].

Here, we will use the hnRNPA1 IDR as a test case to complement the behaviour observed for FUS PLD. We define the IDR of hnRNPA1 as its first 135 residues, 11.9% of which carry a formal charge, and which has been shown to phase-separate *in vitro* [76]. We used a genetic-algorithm-driven evolution of the same fitness function as in the case of FUS; the genetic-algorithm approach results in an increase of fitness whilst maintaining the population diversity [Fig 4A], and the increased fitness function is again a successful proxy for the upper critical solution temperature [Fig 4B]. We also note that in this case, over the three repeats of the genetic-algorithm runs, mutation of all residues was attempted at least once by the genetic algorithm, giving us good coverage of the entire sequence.

The change in amino-acid composition upon applying the genetic algorithm to hnRNPA1 is qualitatively different from the hydrophobicity-driven case of FUS, as hydrophilic/charged residues are not lost and hydrophobic residues appear, statistically replacing some of the highly abundant amino acids of intermediate hydrophobicity [Fig 4C], indicating that the driving force for phase separation may be different from the case of FUS-PLD.

To investigate this further, we have analysed the initial and final populations in the genetic-algorithm runs. As in the case of the PLD of FUS, the hydrophobic attractiveness and the average size of amino acids of the IDR of hnRNPA1 increase to raise the propensity for LLPS [Fig 4D(i)–4D(iii)], and as in the case of FUS, the per-residue effects of hydrophobic attractiveness and amino-acid size are largely, but not fully, correlated.

**Charge patterning and hydrophobicity can co-evolve** . . . Additionally, there is a substantial difference in terms of charge patterning over the course of the genetic-algorithm run [Fig 4D(iv) and 4D(v)]. Charges are both created and lost across the sequence, but not uniformly so. We show in Fig 4E(i) two measures of the charge patterning, the net charge of a protein chain, and the sequence charge decoration (SCD), defined as [87]

$$Q_{\text{SCD}} = \frac{1}{N} \sum_{m=2}^{N} \sum_{n=1}^{m-1} q_m q_n (m-n)^{1/2}, \tag{1}$$

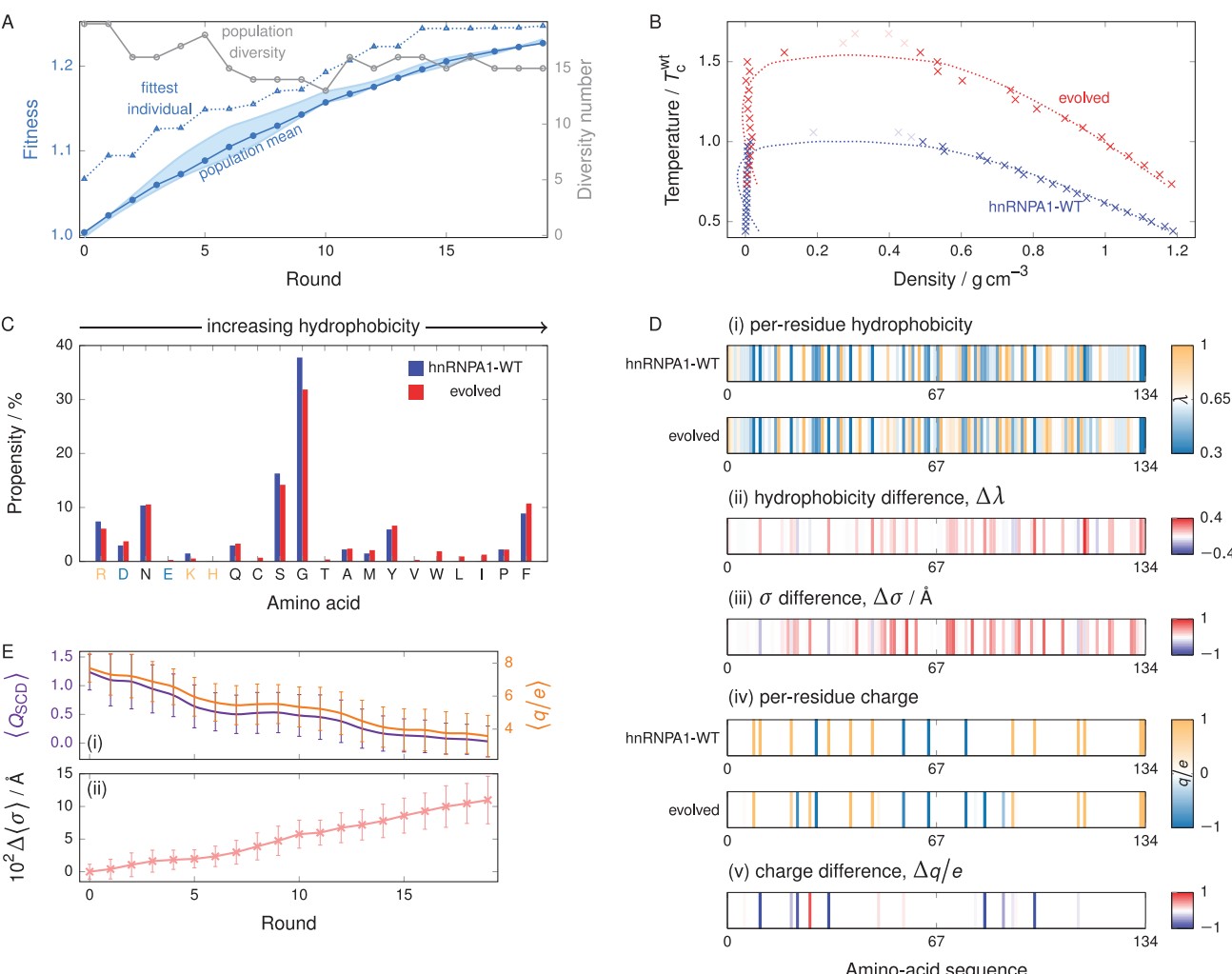

**Fig 4. Evolution of hnRNPA1 IDR. (A)** Typical GA progression for hnRNPA1. The fitness function [Eq (2)] was evaluated at $T = 0.57\,T_c^{wt}$, and increases by $\sim 20\%$ over 20 rounds, while maintaining population diversity. The shaded area corresponds to the range of values of the mean fitness obtained from 3 independent genetic-algorithm runs. **(B)** Comparison of representative phase diagrams before and after genetic-algorithm runs, showing a $\sim 50\%$ increase in critical temperature. Dotted lines are fits, and greyed-out points lie above the critical point, as detailed in Section S1 Text (Sec. S1-b). **(C)** Amino-acid composition before and after using a genetic algorithm to increase the phase diagram width for hnRNPA1, revealing the appearance of hydrophobic amino acids, while no bias against charged or polar amino acids is observed. Amino acids are plotted in order of increasing hydrophobicity [see Table A in S1 Text]. Positively charged amino acids are indicated in light orange and negatively ones in light blue. **(D)** Map of (i) the per-residue hydrophobicity along the sequence, (ii) the change from WT hnRNPA1 to the evolved protein, (iii) the per-residue change $\sigma$ value and (iv), (v) analogous maps for the charge. The data for the evolved protein are averaged over the entire population at the end of 3 independent genetic-algorithm runs. Partial charges reflect only partial carriage in the population. Some charges have appeared and some have disappeared; the overall balance is towards charge neutralisation. **(E)** (i) The sequence charge decoration (SCD) and the charge number ($q/e$) of the population decrease over the course of the genetic-algorithm run, indicating an evolutionary charge neutralisation. (ii) The average $\sigma$ value of the amino acids in the population increases over the course of the genetic-algorithm run. Error bars are standard deviations of the averaged $\sigma$ value of individual sequences with respect to the population.

where $q_i$ is the formal charge number of residue $i$ and $N$ is the length of the amino-acid sequence. $Q_{SCD}$ has been shown to anti-correlate with the upper critical solution temperature of an IDR [45]. These two parameters show a virtually identical evolution through the genetic-algorithm run, which indicates that, in this case, the decrease in charge separation as measured by $Q_{SCD}$ results from a net decrease in the overall charge of the protein. Specifically, this arises from the creation of a larger number of negative than positive charges.

A considerable amount of work has already been done in the context of the role of charge patterning [39, 44, 88–91]. Although it is perhaps not overly surprising that a more even distribution of positive and negative charges allows for the largest number of attractive interactions, which in turn drives the formation of liquid-like phases, we show below that the precise nature in which sequences evolve depends on the medium in which the proteins of interest exist.

The local gradient in sequence space around hnRNPA1-WT has components in both hydrophobicity and charge redistribution. However, in the literature, hnRNPA1 LLPS is not commonly associated with charges [22, 34]. This prompts the question of how important the factors are in absolute terms. A crude estimate can be obtained from an analysis of the components of the pairwise energy in our simulations, shown in Table 1. In particular, we split up the energy into a 'hydrophobic' (LJ) contribution and a Coulomb (electrostatic) contribution. Both components become more favourable over the course of a genetic-algorithm run, indicating that the sequence-space gradient towards higher $T_c$ encompasses both charge rearrangement and hydrophobicity. These effects operate in parallel, but the hydrophobicity component contributes considerably more to the attractive energy in absolute terms.

. . . **but need not necessarily, even in charge-rich sequences.** To check the applicability of the two mechanisms driving the evolution of the capacity to undergo LLPS that we have identified to other protein sequences, we have also investigated an IDR of the protein LAF1, which is a DDX3-family RNA-helicase found enriched in *C. elegans* P-granules, in which it drives phase separation. The IDR we have focussed on has been shown to be both necessary and sufficient for LLPS [92]. It contains a significant proportion of charged amino acids, with 22.4% of its 168 residues carrying a formal charge. This IDR has also been shown to phase-separate in simulations of the CG model used here [35], and a recent study [39] combining CG simulations, all-atom simulations and turbidity assays has identified a sticky hydrophobic stretch as well as tyrosine and arginine residues to be involved in LAF1 LLPS. Additionally, it has been suggested [39] that the even distribution of charges across the sequence may suggest that charge patterning is a controlling determinant of LLPS. Simulations and *in vivo* experiments have been carried out in corroboration of this hypothesis [39].

In order to compare the behaviour of LAF1 to the two cases already considered, we have evolved its sequence using the same genetic algorithm. As before, we have computed the phase diagram at the end of the genetic-algorithm run [Fig 5B]. Although the simulations are slower with this system and finite-size effects are more pronounced, the genetic algorithm with this fitness function can successfully increase the critical solution temperature [Fig 5A and 5B]. Since the higher computational cost restricted us to only one shorter run of the genetic

**Table 1. Changes in contributions to the average pairwise energy per bead between the WT and an evolved sequence of FUS, hnRNPA1 and LAF1.** Standard deviations for the simulation averages are given in brackets and apply to the least significant figure. For LAF1, the evolved population is diverse in terms of these changes, and two representative examples are shown, labelled *[a]* (fitness 1.34) and *[b]* (fitness 1.28). For FUS, sequence evolution results in a change almost exclusively to the hydrophobic part of the pairwise energy. For hnRNPA1 and LAF1*[a]*, both Coulomb and hydrophobic interactions are more favourable in the evolved sequences, but hydrophobic interactions contribute more in absolute terms. For LAF1*[b]*, the Coulomb energy is less favourable in the evolved sequence. All data presented here are obtained at a simulation temperature of 200 K, corresponding to 0.8 $T_c$(FUS). While the overall average energies themselves depend significantly on temperature, the differences between WT and evolved sequence energies are largely independent of temperature in the range of interest.

| System | $10^3 \Delta E_{coul}$/kcal mol$^{-1}$ | $10 \Delta E_{LJ}$/kcal mol$^{-1}$ |
|---|---|---|
| FUS | 2.2(2) | −4.69(1) |
| hnRNPA1 | −34(1) | −6.1(1) |
| LAF1*[a]* | −11.5(5) | −2.11(3) |
| LAF1*[b]* | 3.4(4) | −1.04(3) |

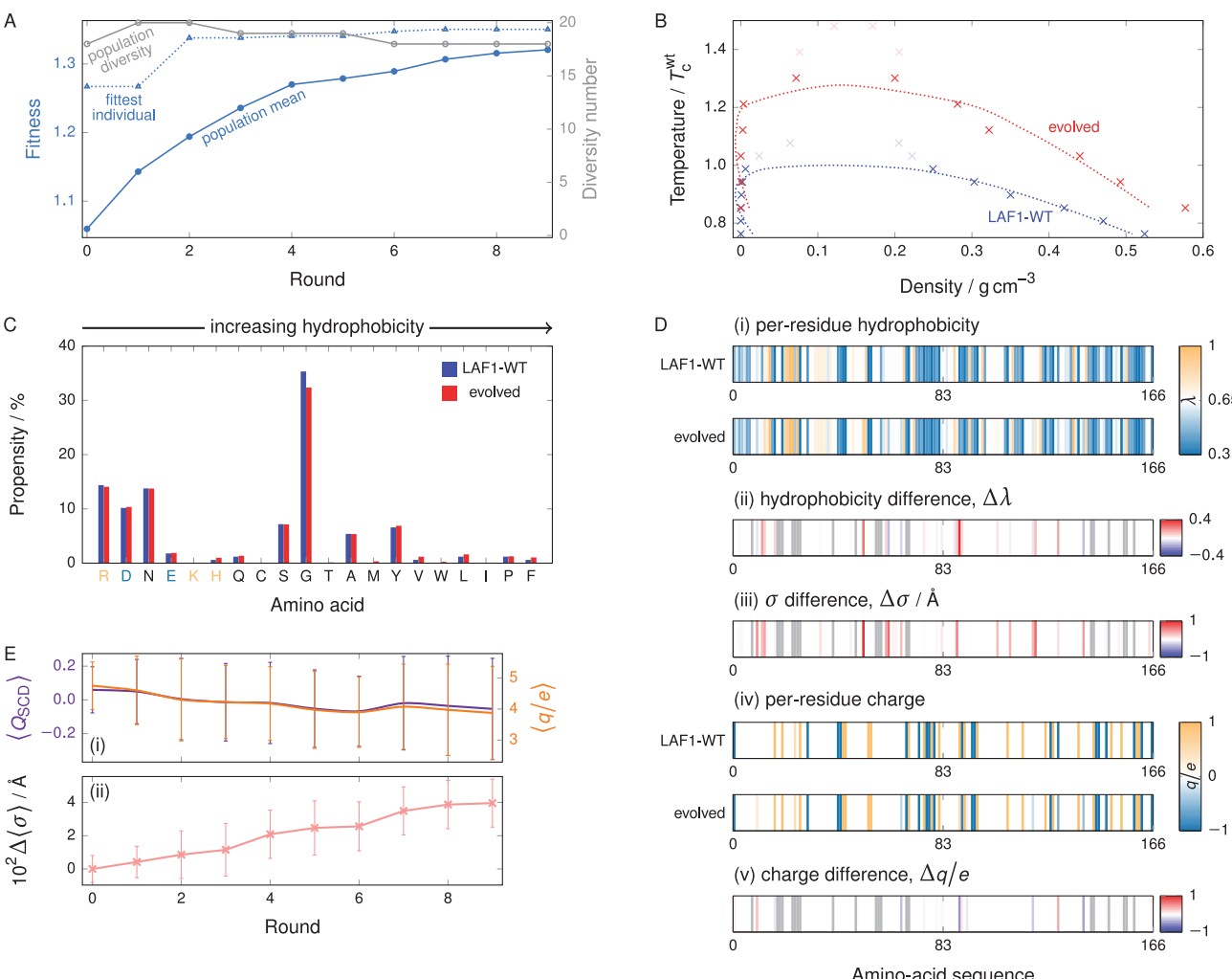

**Fig 5. Evolution of LAF1 IDR. (A)** Typical GA progression for LAF1. The fitness function [Eq (2)] was evaluated at $T = 0.85\,T_c^{wt}$, and increases by $\sim 30\%$ over 9 rounds, while maintaining population diversity. To account for thick interfaces, the simulation box was doubled in all directions compared to simulations of FUS and hnRNPA1. **(B)** Comparison of representative phase diagrams before and after genetic-algorithm runs. Although the phase diagram close to the critical point is especially difficult to equilibrate because of interfacial effects in this system, data points at lower temperatures suggest that the critical temperature increases by $\sim 30\%$ by the end of the GA optimisation. **(C)** Amino-acid composition before and after genetic-algorithm runs targeting an increase in $T_c$ for LAF1. There is a slight general increase in hydrophobicity, whilst hydrophilic and charged residues are largely conserved. Amino acids are plotted in order of increasing hydrophobicity [see Table A in S1 Text]. Positively charged amino acids are indicated in light orange and negatively ones in light blue. **(D)** Map of (i) the per-residue hydrophobicity along the sequence, (ii) the change from WT LAF1 to the evolved protein, (iii) the per-residue change $\sigma$ value and (iv), (v) analogous maps for the charge. Data are shown for one genetic-algorithm run, and those residues where no change was attempted by the genetic algorithm are shown in light grey. Partial charges reflect only partial carriage in the population. There is a slight overall increase in hydrophobicity across the sequence, and there are more charges lost than created during the course of the genetic-algorithm runs. As opposed to the hydrophobicity, the $\sigma$ values increase for almost all those residues that were changed by the genetic algorithm. More charges are lost than created during the course of the genetic-algorithm runs. **(E)** (i) The sequence charge decoration (SCD) and the charge number ($q/e$) of the population decrease slightly over the course of the genetic-algorithm run. (ii) The average $\sigma$ value of the amino acids in the population increases over the course of the genetic-algorithm run. Error bars are standard deviations of the averaged $\sigma$ value of individual sequences with respect to the population.

algorithm, there was incomplete coverage of the sequence with mutations, in contrast to FUS and hnRNPA1. While this is not an issue for interpreting our results, since complete coverage is not essential to observe the trends we look for, we highlight those residues which were not touched by the genetic algorithm for clarity [Fig 5D]. Compared to both FUS and hnRNPA1, the composition of the resulting evolved sequence population is less significantly changed [Fig 5C], although this is consistent with the fact that the fitness function increases more slowly

and the overall critical solution temperature is only ∼30% higher than the wild type in the simulations considered (compared to 65% and 50% for FUS and hnRNPA1, respectively). Nevertheless, there is a limited increase in hydrophobicity [Fig 5D and 5E], with no region particularly favoured in terms of increased hydrophobicity, even though the amino acids early in the sequence (i.e. nearer the N-terminus) are on average more hydrophobic than later ones. The change in the range of hydrophobic interactions, however, as quantified by the average $\sigma$ values [Fig 5E], is more significant ($\Delta\sigma = 0.0396$ Å over 10 rounds). This is comparable to the level of change we observed in FUS ($\Delta\sigma = 0.0322$ Å after the first 10 rounds). In particular, almost all changes made to the sequence by the genetic algorithm lead to larger amino-acid sizes, even though in terms of hydrophobic attractiveness their effects are much more varied. This leads us to speculate that the extended range of attractive interactions may be the dominant factor in driving the evolution of hydrophobic interactions in this case, rather than the $\lambda$ values, which change less.

The change in charge [Fig 5D(v)] is also relatively modest, and mainly entails the loss of existing net charge [Fig 5E(i)]. This is consistent with charge segregation, as quantified by the sequence charge decoration parameter, also shown in Fig 5E(i), which similarly decreases over the course of the genetic-algorithm run, but whose decrease is significantly less pronounced than in the case of hnRNPA1.

Similarly to the case of hnRNPA1, we show in Table 1 the change in the components of the average interaction energy for LAF1 before and after running the genetic algorithm. However, in the case of LAF1, within the final evolved population, different sequences score rather differently in this analysis. Values for two representative evolved variants, termed LAF1*[a]* and LAF1*[b]*, are shown in the table. We have chosen these specific variants as examples of sequences with similar (high) fitness values, but very different contributions to the energy. In particular, LAF1*[a]* behaves similarly to hnRNPA1, with both hydrophobic and Coulomb interactions interactions more favourable in the evolved sequence than in the wild-type, whilst in LAF1*[b]*, the Coulomb energy actually becomes less favourable. Since the genetic algorithm produces sequences in which the Coulomb attractions are enhanced as well as sequences in which they are weakened, charge patterning does not appear to act as an evolutionary driving force in LAF1. Based on these data we conclude that the evolution of the phase behaviour of the LAF1-IDR is primarily driven by hydrophobicity, and in particular by the size of the amino acids in the sequence. Moreover, this result illustrates a useful advantage of the use of genetic algorithms: when phase behaviour can be enhanced in different ways, this can readily be observed, illustrating that the potential energy surface in sequence space in cases such as this has a number of seemingly degenerate or nearly degenerate states.

## Sequence evolution depends on the composition of the medium

Inside cells, proteins are never isolated, and LLPS in multi-component systems can be significantly different from that in single-component ones [26, 93–95]. It is therefore instructive to examine how genetic-algorithm driving behaviour changes upon the addition of a second component to the medium. To this end, we replace one eighth of the chains in the system with a poly-U RNA chain of the same length as the respective protein of interest. Since cation–π interactions are known to be important for RNA–protein interactions [96], we employ the cation–π model [58], a reparameterised version of the coarse-grained model we have used so far, alongside RNA–protein interaction parameters from Regy and co-workers [59]. We discuss this model further in S1 Text (Sec. S4). The goal of these tests is simply to determine if our genetic algorithm is sensitive enough to evolve the amino-acid sequence of a given protein so as intentionally to modulate its phase-separating behaviour while taking into account the

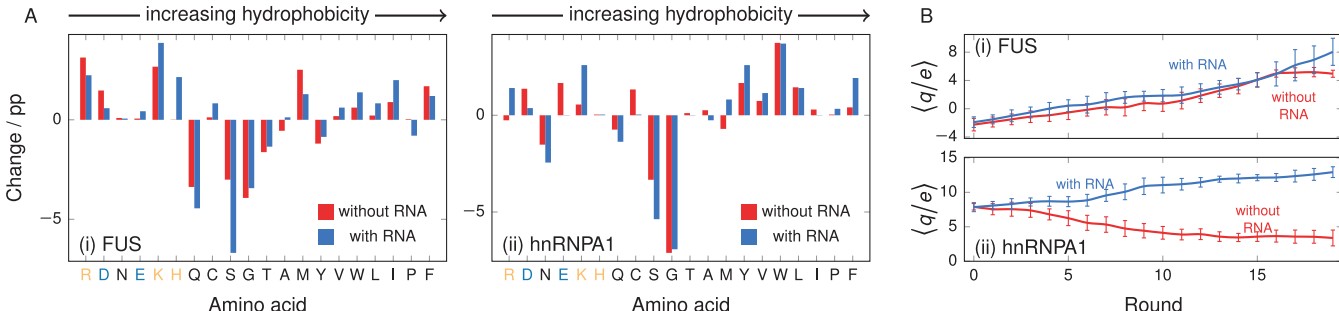

**Fig 6. Changing the composition of the medium. (A)** Percentage-point difference in amino-acid composition after increasing the phase diagram width for (i) FUS and (ii) hnRNPA1. Amino acids are plotted in order of increasing hydrophobicity [see Table A in S1 Text]. Positively charged amino acids are indicated in light orange and negatively charged ones in light blue. The 'with RNA' series corresponds to a system where 1/8 of the chains of the system are replaced with poly-U RNA of the same length as the IDR. **(B)** Charge content as a function of genetic-algorithm round for (i) FUS and (ii) hnRNPA1. Error bars give standard deviations for the population at each round. The addition of RNA changes the evolution behaviour significantly, particularly in the case of hnRNPA1, favouring higher charge content with the creation of new positive charges, illustrating that the evolutionary driving force depends not only on the initial sequence of the protein, but also on the medium around it.

condensate composition. A poly-U RNA molecule has a high negative charge density and significant scope for non-bonded interactions, which gives it biophysical properties significantly different from the protein it is replacing, making this a good initial test case for more complex multi-component systems.

We show in Fig 6A the change in amino-acid composition at the end of the genetic-algorithm run in systems with and without the additional RNA component, contrasted for (i) FUS and (ii) hnRNPA1. For both proteins, the amino-acid composition change in the presence of a RNA component is different from the single-component case. While the precise changes incurred are likely dependent on the particular model parameters, it is noteworthy that the genetic algorithm can fine-tune composition to enhance phase separation in different ways, depending on the composition of the medium. This applies beyond changes to compensate simply for the introduced charge: in the case of FUS, the overall sequence charge is not drastically affected by the presence of RNA [Fig 6B(i)]; indeed, as can be seen from Fig 6A(i), the creation of more lysine (K) residues is largely offset by creating fewer arginine (R) residues, which have the same charge, and so the net charge increases to a broadly similar extent both with and without RNA present. By contrast, we can observe in Fig 6B(ii) that in the case of hnRNPA1, the addition of RNA affects the evolution in a very different way from that of FUS; namely, the presence of RNA leads to the increase in net positive charge, as opposed to its reduction by the genetic-algorithm run in the absence of RNA. These results illustrate that with the genetic-algorithm approach, we can not only probe the evolutionary driving forces resulting from changing the composition of the medium in which LLPS occurs, but we can also gain insight into how such driving forces depend on the sequence of the protein of interest. In other words, evolutionary driving forces due to the starting sequence and the medium are coupled and evolved together by the genetic algorithm.

Interestingly, all IDRs studied in this work are derived from proteins which in their full-length variants bind to RNA. RNA has widely been studied in the context of LLPS, and can, depending on its concentration, both promote and inhibit protein phase separation [76], potentially even resulting in re-entrant phase behaviour [97, 98]. Therefore, we suggest that extending these preliminary trials on multi-component systems with genetic algorithms could provide insights into the mechanisms by which IDR-containing proteins and RNA might recruit each other.

## Experimental validation

In order to probe the validity of the trends seen in our genetic-algorithm calculations, and hence the interpretation of the genetic-algorithm predictions, we have computed a series of phase diagrams corresponding to experimental modifications of the IDP of hnRNPA1 as studied in [23], whose sequences are given in S1 Text (Sec. S8). We computed these with two coarse-grained potentials, namely the Kim–Hummer-style parameterisation of the coarse-grained potential of [35], and the cation–π reparameterisation of the 'HPS' potential introduced by [58] We show the phase diagrams for these sequences in Fig 7A. From the simulation data points, we fit the data to Eqns (S6) and (S7) of S1 Text, with low-density predictions truncated to zero. From these, we obtain an estimate of the critical temperature in each case. Due to coarse-graining, the absolute temperature scale is of course not directly comparable to experiment; however, if the models are good, one might expect the temperature relative to the critical point to be meaningful. For each sequence, we have therefore estimated the experimental critical temperature from the experimental phase diagrams of [23], and we show a correlation plot of simulation and experimental data in Fig 7B. The linear fits have very reasonable adjusted squared sums of residuals and Pearson correlation coefficients [see Fig 7], and the coefficient of the linear term has $p$ values of 0.0007 and 0.012, respectively, indicating the statistical significance of the predictor (relative to the null hypothesis of a constant line). The Pearson coefficient for the predictions of the two models against each other is 0.74, and although the majority of data points agree rather well, there are some outliers, particularly for

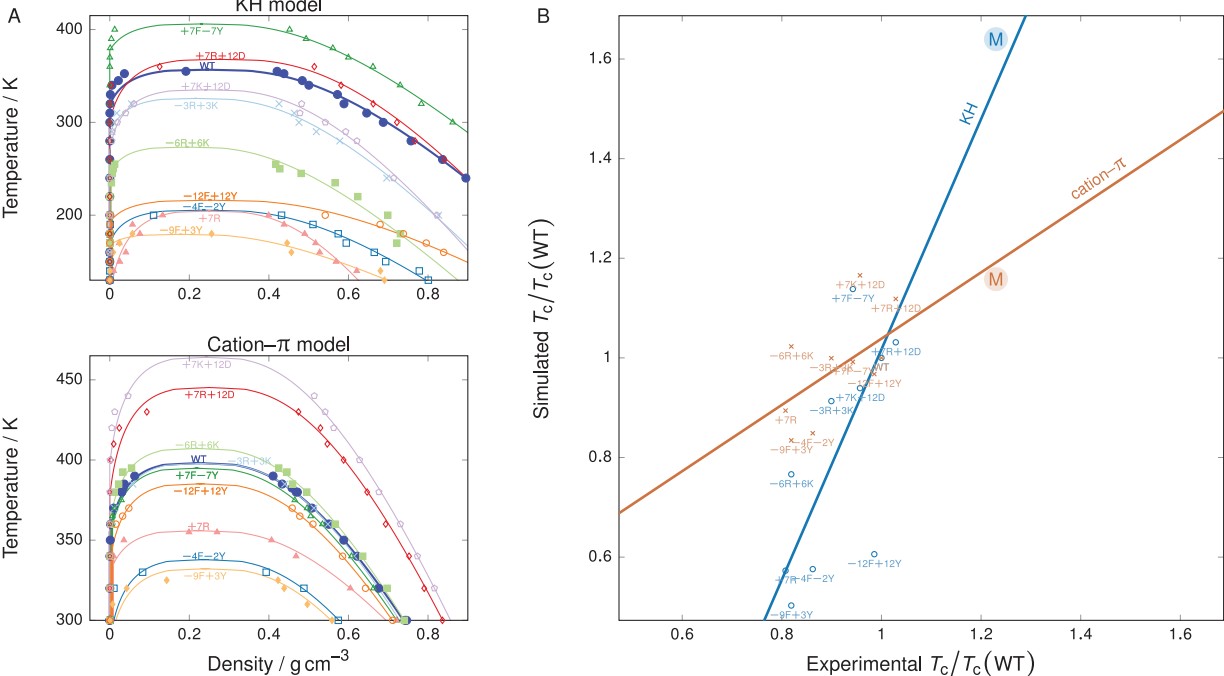

**Fig 7. Comparison of model predictions with experiment.** In **(A)**, we show phase diagrams obtained with two coarse-grained models for a variety of modifications of the hnRNPA1 sequence. Symbols correspond to simulation results. Solid lines are obtained by the fitting procedure described in S1 Text (Sec. S1-b). In **(B)**, we show the correlation plot between simulation and experimental data, alongside a linear fit to the data points. The adjusted squared sum of residuals is $R^2 = 0.96$ for the KH model and $R^2 = 0.99$ for the cation–π model, with $p$ values of 0.0007 and 0.012, respectively. The Pearson correlation coefficients are 0.66 (KH) and 0.64 (cation–π). The point labelled 'M', which was not included when computing the linear fit or the correlation coefficients, corresponds to the sequence with the largest critical point obtained in our genetic-algorithm run (see S1 Text (Sec. S7)); since there are no experimental data available for this specific sequence, we have estimated the experimental critical temperature using the ABSINTH model.

those amino-acid sequences with significant lysine (K) or tyrosine (Y) content. The good positive correlation between the predictions of the sequence-dependent coarse-grained models we have used and the experimental results suggests that we can use these models with success to study broad trends. However, the agreement is by no means perfect for either of these simplified models, demonstrating that there is scope for improving coarse-grained potentials to describe LLPS more accurately.

Next, we probe our final predictions for FUS, hnRNPA1 and LAF1 further by using the more realistic ABSINTH model [49] to estimate the θ temperature of these proteins. The θ temperature of single-molecule coil-to-globule transitions is a well-established proxy for the critical solution temperature of IDR solutions [38, 74, 99]. It is defined as the temperature at which, for a single protein, there is a coiling transition as evidenced by a sudden change in its radius of gyration as a function of temperature [51]. Linking the single-molecule θ temperature to the critical solution temperature is possible for homotypic LLPS when the driving forces for the single-molecule coil-to-globule transitions are similar to those stabilising the phase transition. Therefore, to validate our results, we calculate the θ temperature firstly for WT-FUS and two of our evolved FUS PLD variants by performing all-atom Monte Carlo simulations in implicit solvent using the ABSINTH framework [36, 49], as implemented in the CAMPARI code [50]. The advantage of doing this is that ABSINTH is currently one of the most reliable modelling approaches to produce experimentally consistent conformational ensembles of IDRs [100–103] and to predict IDR critical solution parameters in agreement with experiment [22, 51]. The success of ABSINTH is anchored in its extensive experimental validation and refinement, and use of experimentally derived reference solvation free energies. Reassuringly, ABSINTH ranks both our FUS variants and our LAF1 variants from low to high θ temperatures in the same order as the coarse-grained models.

For hnRNPA1, we can not only compute the relative ordering of θ temperatures with ABSINTH, but we can also compute approximate critical points for the wild-type hnRNPA1 and some of its analogues for which experimental results are available [23]. We have computed these for the wild type and the +7R and +7F−7Y analogue, and these estimates agree very well with the experimental critical points determined in [23]: for WT-hnRNPA1, the experimental critical temperature is ∼ 348 K and the ABSINTH estimate is ∼ 345 K; for the +7R analogue, they are ∼ 280 K and ∼ 275 K, and for the +7F−7Y analogue, they are ∼ 328 K and ∼ 325 K. We have also determined the analogous result for the sequence with the largest critical point obtained in the genetic-algorithm run (see S1 Text (Sec. S7)), and we show it alongside the experimental results in Fig 7B. For both coarse-grained potentials we compare, this point falls very close indeed to the prediction of the linear fit from experimental results, suggesting that the simple coarse-grained potentials are sufficiently powerful to obtain qualitative insight into the phase behaviour of intrinsically disordered proteins.

## Discussion

In this work, we have proposed an efficient computational method to evolve naturally occurring phase-separating protein sequences. Evolving such sequences can provide insight into which sequence features drive LLPS, both when the proteins are in pure form and when they form part of a multi-component mixture; moreover, our approach could also be extended to design experimentally testable amino-acid sequence mutations which either inhibit or promote the LLPS of protein solutions. Our approach combines state-of-the-art molecular simulations of protein condensates, where each protein is described at the single-amino-acid resolution, with a genetic algorithm grounded in a new fitness function—the difference in composition densities of the protein-poor and protein-rich phases at constant volume—which

is both a good proxy for the critical solution temperature and computationally far more tractable to obtain than the critical temperature itself. We have shown that such a simple and computationally inexpensive fitness function is sufficient to evolve the amino-acid sequences of naturally occurring proteins and to shift their phase behaviour in the direction we choose. Moreover, by analysing the effects of small changes to naturally occurring amino-acid sequences, we can draw conclusions about the molecular origins of the local gradient in amino-acid sequence space, adding to conventional analyses of driving forces which usually focus on binding energies. Indeed, an important finding of our work is that we have demonstrated how a genetic-algorithm framework that can alter the LLPS behaviour of proteins also enables us to probe the gradient in amino-acid sequence space directly. This can help us both to extend interpretations of why proteins that drive the formation of condensates might have evolved as they have and to gain greater control over intracellular LLPS.

We have coupled our genetic-algorithm approach to the coarse-grained model of [35], which is one of the best simple models currently available for probing the phase behaviour of protein solutions. This model has been validated against the single-molecule experimental radii of gyration of a wide range of IDRs [35], is residue-specific, has been shown to reproduce well the experimental phase behaviour of various proteins under different conditions [21, 39, 47, 104, 105], and is computationally sufficiently inexpensive that it affords the determination of bulk LLPS properties for many sequences. Furthermore, the model accounts for key physicochemical aspects that determine the phase behaviour of proteins, such as the charge, size, relative hydrophobicity and flexibility of amino acids. However, as is the case with any coarse-grained model, it is still approximate and averages out other effects, in this case especially the specific contribution of π–π interactions, polarisation effects that give rise to cation–π interactions and the explicit role of water and ions in solution. To benchmark the robustness of the protein model, we have therefore repeated our simulations with another coarse-grained potential that can account for cation–π interactions [58] and protein–RNA interactions [59]. The two models predict the same behaviour on evolution, as we discuss in S1 Text (Sec. S4), which suggests that the trends we have outlined are not especially sensitive to the choice of model. The evolution of protein amino-acid sequences is a computationally expensive process, and is only made feasible by the choice of a suitably coarse-grained potential. However, it is not immediately clear that such simplified models, which were largely validated against single-molecule experiments, are predictive in the context of LLPS. We have therefore validated the model predictions against experimental results [23] and an all-atom potential [49, 50]. Despite the fact that the models' predictions are not quantitative, the trends in critical temperature predicted by the models we have used correlate well with experimental results, suggesting that such simple coarse-grained potentials are sufficiently powerful to obtain qualitative insight into the physical driving forces governing the phase behaviour of intrinsically disordered proteins. Moreover, the genetic algorithm we have introduced can of course straightforwardly be used with protein models of higher resolution and accuracy as they are developed, provided that sufficient computational resources are available.

Although the fine details of the phase behaviour we have observed may be model-specific, we have nevertheless shown two distinct driving regimes for enhancing or inhibiting LLPS to exist, namely hydrophobicity—including both the strength and range of relevant interactions—and charge patterning. In sequences such as the PLD of FUS, only one may be in operation, whilst in others, such as hnRNPA1-IDR, they may co-evolve, implying that both driving forces can contribute to LLPS simultaneously. Furthermore, in the hydrophobic driving regime in the case of the PLD of FUS, we have shown that there is a patterning length scale of 2–3 amino acids, which one can interpret in the context of the stickers-and-spacers model of

proteins. Intriguingly, we have shown that although LAF1-IDR is charge-rich, charge patterning does not appear to co-evolve with hydrophobicity. In all cases studied, LLPS is facilitated by an increase in the mean size of the amino-acid residues of the proteins, which results in a more structured protein-rich phase, which in turn can favour condensate formation. It would be especially interesting to investigate in future work whether such a driving force for phase separation is more universal than might previously have been thought. Finally, we have demonstrated that the genetic-algorithm approach is successful at evolving sequences in the presence of other species in the medium; here, we have focussed on simple RNA molecules as a proof of concept. Significant changes to the gradient in sequence space are observed when another species is introduced into the system, indicating that our method is also suitable for investigating the co-evolution of proteins and for studying biologically relevant mixtures of different species. Since the effect of sequence modifications on the phase behaviour of many-component mixtures is much less intuitive to predict manually than it is in one-component systems, the ability to guide phase behaviour algorithmically is especially attractive.

In summary, we have presented a powerful framework for systematically modulating the LLPS of proteins by evolving their amino-acid sequences. We have shown that the approach is able to provide direct insight into the nature of LLPS in protein solutions, demonstrating both which fundamental driving forces are in operation as well as providing specific guidance into the kinds of mutation that may help promote or inhibit LLPS in practical applications. Recently, several databases of proteins exhibiting LLPS have been assembled [106–108], providing an excellent starting point for determining and contrasting the driving forces governing phase separation in very different systems. We have already drawn useful conclusions from the application of our approach to specific cases, contributing a significant piece of the puzzle towards a fuller understanding of the physical driving forces behind LLPS. As ever more accurate force fields of proteins in solution are developed, this approach promises to be particularly fruitful in furthering our understanding of the regulation of LLPS in biology, as well as representing a first step towards future engineering of phase-separating sequences.

## Methods

In S1 Text, we describe the coarse-grained potential, provide further details about the computational methods used, provide further analysis and additional supporting results, and provide the sequences of the proteins studied.

### Simulation methods

We performed molecular dynamics simulations of a coarse-grained implicit-solvent model of proteins [35] in which each amino acid is represented as a bead. Neighbouring amino acids in a protein chain are connected by harmonic springs, while other beads interact with one another with a hydrophobicity-scaled Lennard-Jones (LJ) potential and a Debye–Hückel electrostatic potential. The model is discussed in more detail in S1 Text (Sec. S1-a), and the simulation methods in S1 Text (Sec. S1-b). We have also used a further coarse-grained model for validation [58] and for protein–RNA simulations [59].

### Genetic algorithms

Genetic algorithms optimise properties of a system in ways inspired by biological adaptation of populations [109–111]. All numerical parameters listed below were chosen to balance the need for high evolution speed due to an expensive fitness function and that sufficient diversity in the population be maintained to avoid premature convergence.

1. We define a chromosome of length $n$, $\boldsymbol{x}_i = (x_{i1}, x_{i2}, \ldots, x_{in})$, where in our case $x_{ij}$ is an amino acid. A set of $N$ chromosomes defines an initial population $U_0 = \{\boldsymbol{x}_1, \boldsymbol{x}_2, \ldots, \boldsymbol{x}_N\}$, where $U_t$ denotes the population of a given round $t$. The starting population in our case corresponds to mutated versions of the WT $\boldsymbol{x}_{\mathrm{WT}}$ with a certain mutation rate, i.e. the frequency at which an amino acid is exchanged for a random one picked from the natural set of 20. We use a rate of 0.01 in this work. A scalar fitness function $f(\boldsymbol{x})$ denotes the property being optimised. We use the width of the phase diagram at a fixed temperature as a proxy for the critical solution temperature, and define the fitness function as

$$f(\boldsymbol{x}) = \frac{\rho_{\mathrm{l}}(\boldsymbol{x}) - \rho_{\mathrm{v}}(\boldsymbol{x})}{\rho_{\mathrm{l}}(\boldsymbol{x}_{\mathrm{WT}}) - \rho_{\mathrm{v}}(\boldsymbol{x}_{\mathrm{WT}})} \quad , \tag{2}$$

where $\rho_{\mathrm{l}}(\boldsymbol{x})$ is the average density of the 'liquid-like' protein-rich phase of sequence $\boldsymbol{x}$ and $\rho_{\mathrm{v}}(\boldsymbol{x})$ is the analogue for the 'vapour-like' protein-poor phase. We refer to individuals with high fitness value as 'strong' and to those with low fitness value as 'weak'. In our case, $N = 20$. For genetic-algorithm runs in which the target is to reduce the critical solution temperature, we use as the fitness function the reciprocal of $f(\boldsymbol{x})$.

2. In each round $t$, we choose $N_{\mathrm{par}} = 8$ parents $P$ from the population, $P \subset U_t$. To achieve this, we use tournament selection [111]: We first define $N_{\mathrm{par}}$ tournaments $T_i$. Each tournament is a randomly drawn subset of $N_{\mathrm{tour}}$ elements from $U_t$. The fittest sequence from each tournament becomes one of the parents. The tournament size $N_{\mathrm{tour}}$ is therefore a direct scaling parameter governing the selection pressure. For our purposes, we have found that $N_{\mathrm{tour}} = 5$ works well.

3. The parents are randomly divided into pairs $(\boldsymbol{a}, \boldsymbol{b})$ (where $\boldsymbol{a} \in P$ and $\boldsymbol{b} \in P$), and crossed over. Here, we swap sequences after a randomly chosen position $k \in [1, n]$ in the sequences, such that

$$(a_i, b_i) = \begin{cases} (a_i, b_i) & \text{if } i \leq k, \\ (b_i, a_i) & \text{otherwise.} \end{cases} \tag{3}$$

4. Another round of random mutations with the same mutation rate is then performed to cover previously unrepresented areas of sequence space.

5. The result from steps 3 and 4 is a set of children $C$, whose fitness is then evaluated. Children replace some chromosomes in the population. As fitness functions are relatively expensive to compute for our system, we use weak-population replacement [109], a greedy algorithm that can achieve rapid population evolution. Sequentially, each child $\boldsymbol{c}_i \in C$ is compared to the weakest individual in the population, $\boldsymbol{x}_{\mathrm{weak}} \in U_t$. If $f(\boldsymbol{c}_i) > f(\boldsymbol{x}_{\mathrm{weak}})$, then $\boldsymbol{c}_i$ replaces $\boldsymbol{x}_{\mathrm{weak}}$. The weakest individual may also be a previously inserted child.

Parallelisation can speed up genetic-algorithm progression [112]. We use a simple master–slave approach since asynchronous schemes are not well suited to small populations; because all simulations are run for the same amount of wall-clock time, the overhead of the simple genetic-algorithm parallelisation employed is small compared to the duration of individual simulations.

## Supporting information

**S1 Text. Computational details, ancillary discussion and listings of amino-acid sequences.**
Includes Table A. **Table A**: **Table of amino acids**. The 20 naturally occurring amino acids
with their one- and three-letter codes, alongside their charges. Amino acids marked with a '⋆'
are aromatic. The last columns give the $\lambda$- and $\sigma$-parameters which define the hydrophobicity
scale.
(PDF)

**S1 Fig. Direct-coexistence simulations.** (A) Orthographic projections of simulation boxes of
FUS at two temperatures below $T_c$, as indicated, where different colours represent different
protein chains. The box is periodic in all directions. (B) Example of a fitted density profile for
FUS at $T = 0.8\ T_c$. The simulation box along the $z$ direction is split into 25 bins and the average
density is computed within each bin, indicated by blue crosses. Points from the regions where
solid lines are shown were used to compute a fit to a constant for each of the high- and the
low-density phases, taking into account an interface of finite thickness.
(EPS)

**S2 Fig. Hydrophobic sequence optimum.** Phase diagram of the $(Phe)_{163}$ sequence, the same
length as FUS-WT. The critical temperature is approximately nine times that of FUS-WT. The
dotted line is a fit, and greyed-out points lie above the critical point, as detailed in S1 Text (Sec.
S1-b).
(EPS)

**S3 Fig. Dummy genetic algorithm.** A dummy genetic-algorithm progression (orange), con-
trasted with the driven evolution (blue) of FUS phase separation. We remove any driving force
by randomising the selection of parents and replacement in the population, but keeping all
mutation steps.
(EPS)

**S4 Fig. Reducing the critical temperature.** (A) Typical GA progression for FUS where the fit-
ness function *reduces* the upper critical solution temperature. The fitness function [defined as
the reciprocal of Eq (2) of the main manuscript] increases by $\sim 90\%$ over 20 rounds. The fittest
individual can be considerably fitter than the mean. The population diversity, i.e. the number
of distinct sequences present in the overall population of 20, is generally very high. (B) Com-
parison of representative phase diagrams before and after genetic-algorithm runs, confirming
that the fitness function choice was suitable. Dotted lines are fits, and greyed-out points lie
above the critical point, as detailed in S1 Text (Sec. S1-b).
(EPS)

**S5 Fig. Chunk-shuffling example runs.** (A) Chunk shuffling at chunk lengths 1, 5, 10 and 15.
Chunk length 1 lies significantly below the other curves. Shaded areas are standard deviations
from 3 (6 for chunk length 1 and 5) shuffling runs. (B) Chunk shuffling with focus on chunk
lengths around a length scale of 2–3 amino acids. Shaded areas are standard deviations from 3
(6 for chunk lengths 2 and 5) shuffling runs. (C) Hydrophobicity-biassed chunk shuffling, only
allowing exchanges between the top 30% chunks by hydrophobicity. Shaded areas are standard
deviations from 3 shuffling runs.
(EPS)

**S6 Fig. Evolution with the cation–π model.** Results from the application of the genetic
algorithm to (A) FUS and (B) hnRNPA1 using the cation–π model. In each case, panel
(i) shows the change in fitness function and the population diversity as a function of the

genetic-algorithm round, while panel (ii) shows a comparison of the evolved amino-acid propensities for the two sequences when evolved with the model used elsewhere ('KH') and the cation–π reparameterisation, with the wild type shown for reference. Amino acids are plotted in order of increasing hydrophobicity [see Table A in S1 Text]. Positively charged amino acids are indicated in light orange and negatively ones in light blue.
(EPS)

**S7 Fig. Liquid character of phases.** Phase diagram of wild-type hnRNPA1 (in blue) and of the hnRNPA1 analogue obtained at the end of the genetic-algorithm run (in red), alongside diffusion coefficients for the liquid-like phase (green circles for the wild type, green crosses for the evolved sequence, measured along the green abscissa). The densities of the coexisting phases were first determined in direct-coexistence simulations at a range of temperatures. The diffusion coefficient was then computed in a canonical-ensemble simulation at the density corresponding to the liquid-like (high-density) phase. A non-linear relationship exists between the diffusion coefficient and temperature, but there is no indication of a discontinuous change that might indicate a glass transition.
(EPS)

**S1 Data. Raw data, LAMMPS scripts, analysis scripts and sequence listings along genetic-algorithm runs.** The supporting data archive contains the raw data used in the figures; a series of LAMMPS scripts that can be used to generate the simulation data; analysis scripts used to post-process the data; and a full listing of all sequences across all genetic-algorithm runs reported alongside their phase-diagram widths.
(ZIP)

## Acknowledgments

We thank Jeetain Mittal and Gregory L. Dignon for their invaluable help with implementing their sequence-dependent coarse-grained protein model in LAMMPS.

## Author Contributions

**Conceptualization:** Simon M. Lichtinger, Rosana Collepardo-Guevara, Aleks Reinhardt.

**Investigation:** Simon M. Lichtinger, Aleks Reinhardt.

**Methodology:** Simon M. Lichtinger, Rosana Collepardo-Guevara, Aleks Reinhardt.

**Software:** Simon M. Lichtinger, Adiran Garaizar.

**Supervision:** Rosana Collepardo-Guevara, Aleks Reinhardt.

**Visualization:** Simon M. Lichtinger, Aleks Reinhardt.

**Writing – original draft:** Simon M. Lichtinger, Rosana Collepardo-Guevara, Aleks Reinhardt.

**Writing – review & editing:** Simon M. Lichtinger, Rosana Collepardo-Guevara, Aleks Reinhardt.

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
