## [Decision Letter · Decision Letter 0]

17 Jul 2021

Dear Dr Reinhardt,

Thank you very much for submitting your manuscript "Targeted modulation of protein liquid-liquid phase separation by evolution of amino-acid sequence" for consideration at PLOS Computational Biology. As with all papers reviewed by the journal, your manuscript was reviewed by members of the editorial board and by several independent reviewers. The reviewers appreciated the attention to an important topic. Based on the reviews, we are likely to accept this manuscript for publication, providing that you modify the manuscript according to the review recommendations.

Sincerely,

Damiano Piovesan

Guest Editor

PLOS Computational Biology

Arne Elofsson

Deputy Editor

PLOS Computational Biology

[LINK]

Reviewer's Responses to Questions

**Comments to the Authors:**

Reviewer #1: The authors presented a high-quality and original piece of work aiming to efficiently modulate the liquid-liquid phase separation of proteins by iteratively introducing changes in the protein sequence by a genetic algorithm. The method relies on coarse-graining the protein structure and a simple but effective fitness function, and is able to estimate the phase diagram of the protein. The build-up of the manuscript is logical, the language is clear, the results and methods are detailed and depicted sufficiently. I very much appreciated the use of 3 different protein systems, the comparison of different scoring models and the efforts for the validation. With the incorporation of the mostly minor points listed below, I highly recommend the publication of the article.

Major point:

1. It would be very insightful to compare the genetic algorithm with a dummy algorithm that introduces random mutations (using the same number of rounds or iteration cycles).

Minor points:

1. Fig 2B, Fig 4B, Fig 5B seem to use different colours from the colour scales, as white colour does not appear on the blue-brown-orange and blue-purple-red colour scales, only on the figures. Please show the appropriate colour scales.

2. Statistical significance is unclear in the following sentence: "This pair correlation function exhibits a more pronounced nearest-neighbour maximum, indicating a greater degree of local structure and an increase in the number of nearest-neighbour beads compared to the WT..."  Could you please add the appropriate statistics?

3. In Fig 3A the colour scale of hydrophobicity should match with that of Fig 2B [0,1] in order to avoid interpretation bias.

4. Please add the correlation coefficient to the following sentences of the main text: "Fig. 3(A) shows the results of a glycine scan in sequential chunks of 6 amino acids, projected onto the chunk-averaged hydrophobicities of the WT protein. The curves anti-correlate for most of the sequence…"  & "Our comparison indicates that there is a good positive correlation between the predictions of the sequence-dependent coarse-grained models we have used and the experimental results…"

5. When simulating two-component systems, the RNA:protein stoichiometry was chosen to be 1:7, which I believe is not physiologically relevant. What about choosing a more relevant stoichiometry for an additional GA run?

6. Please add the correlation coefficient for the simulated data using KH and cation-pi models. This would quantify the disagreement between the two models.

7.i. The word amino acid is written as 'amino acids' or 'amino-acids' inconsistently. Please remove/add the hyphens.

7.ii. The same issue was found for 'genetic algorithm' vs 'genetic-algorithm'.

8. I’m very happy to see that the data will be made publicly available. However, it is a pity it is unavailable under the review. Please make sure that the analysis scripts to reproduce the evaluation of the simulations results are also archived with the simulation files.

Reviewer #2: The manuscript by Lichtinger and colleagues presents an interesting computational approach for investigating the effect of sequence modifications in Intrinsically Disordered Regions (IDRs) in three well-studied proteins (FUS, hnRNPA1, LAF1) known to undergo homotypic liquid-liquid phase separation (LLPS) and their LLPS potential. In particular, they propose a simple to compute fitness function to guide a Genetic Algorithm (GA) coupled with coarse-grained molecular dynamics (MD) simulations towards evolving sequences that have desired (increased or decreased) LLPS tendency compared to the wild type.

The work presented is novel, and aims to tackle an important and timely problem. Overall, the manuscript is well written, starting with a brief -yet informative- introduction on the current knowledge in the current knowledge of the driving forces behind LLPS. Then the authors describe a computationally tractable approach to tackle the problem of training a GA to shift the phase behavior of a starting (WT) sequence to a local optimum. This becomes feasible by introducing the diﬀerence in composition densities between the protein-rich and protein-poor phases in an appropriately designed coarse-grained MD simulation of the protein region in question. The authors present results on three IDRs know to drive LLPS in vitro and in vivo, and validate their results against more detailed, all-atom MD simulations. The results presented herein do not identify a single driving force for LLPS but provides some useful insights on parameters other than charge, hydrophobicity and their patterning along sequences that should also be considered when rationally designing mutations for promoting/inhibiting LLPS. In addition, examples where RNAs are introduced in the FUS/hnRNPA1 simulations are presented, aiming to demonstrate the potential of the proposed approach to handle more complex systems.

Below are some points that I believe warrant some attention by the authors. Page numbers refer to the page number of the final PDF file produced in the Editorial Manager system and not the page numbers inserted by the word processing tool.

+ Specific comments

- The manuscript is accompanied by detailed supplementary material. Even though some methodological details (especially those referring to existing methods) are safe to reside in the supplement, I would prefer a large portion of this material to be transferred to the main text (e.g. S4-S9, S12).

- Regarding the results presented in Fig 2A, it is apparent that when FUS-WT is evolved to either high or low Tc, Ser residues seem to decrease (same holds for Gln and Thr even though these changes seem weaker). Could this observation be explained by some positional preferences? Please discuss.

In addition, in Fig 2B it is mentioned that "Trends in hydrophobicity and are largely correlated". It is unclear though if this correlation is somehow quantified. In addition, there exist several clusters of successive residues that seem to not have undergone mutation: doe these correspond to cases where mutations in these positions were not explored by the GA or that such mutants were rejected during the simulation? Are there any particular features of the WT sequences in these stretches? Please discuss.

- Regarding the Glycine-scans presented (pg. 12), a chunk size of 6 AAs was chosen. However, the results shown in Fig 3B are interpreted by the authors that segments of 2-3 successive AAs are crucial in driving LLPS in FUS. Would it be more meaningful if the Glycine-scans were performed with a chunk size of 2 or 3 AAs?

Another question related to Fig 3B: when chunks a shuffled it is not necessary that all residue types are changed - is this taken into account?

- It is unclear what the two evolved variants of LAF1 mentioned in pg. 16 refers to. Please explain how these variants were selected and what is their relevance.

- Under section (V) the authors interpret the results shown in Fig 6A in a descriptive way. They mention wrt FUS that "the creation of more lysine (K) residues is largely offset by creating fewer arginine (R) residues, which have the same charge". However, simple inspection of Fig 6A indicates that net positive charge (especially if His is also taken into account) seems to clearly increase in the presence of RNA (while negative charge decreases). With minor only changes involving negatively charged residues I suspect that there might be some misinterpretation here (or I have missed some crucial detail).

+ Minor comments and typos

- pg. 8. When mentioning "sufficient number of copies": How is this number selected?

- pg. 9. "[(2)]" should better read "[(Equation 2)]. The same also holds for the legend of Fig 1 (B).

- pg. 16. It is unclear to me (a) why the GA progression is slower with this system and (b) why are finite-size effects are more pronounced? The LAF1 IDR is approx the same length compared to the respective FUS domain, so I suspect it is not a matter of length. Could you please elaborate?

- pg. 16. "... the amino acids early in the sequence ...", does it refer to the most N-terminal residues? Please clarify.

- pg. 19. Since the preprint by Bremer et al [ref. 23] is quite new and most readers may not be very familiar with the naming convention of hnRNPA1 analogues (+7R and +7F−7Y), please consider briefly describing them in the text or supplement for clarity.

- pg. 21. "encourage or inhibit" might better read as "promote or inhibit".

- pg. 22. The symbol P refers to denote either the mutation rate (point 1) or the set of parents (point 2) which could be misleading.

- pg. 31. Under S1, in the mutant hnRNPA1 sequences it could be useful to highlight mutations wrt the WT.

Reviewer #3: The paper entitled “Targeted modulation of protein liquid-liquid phase separation by evolution of amino-acid sequence” by Lichtinger et al. addresses an important and challenge topic in the field of LLPS, aiming to understand how single point amino acid variations affect the driving forces in LLPS, modulating it. They followed a novel computational methodology by combining genetic algorithms and sequence-dependent coarse-grained protein models, to evolve amino-acid sequences of phase separating IDPs. The authors performed a deep analysis with a very precise and rigorous methodology on the different sequence properties as hydrophobicity and residue charge in the wild type and the evolved sequences of PLD of FUS, hnRNA and LAF1, well-studied proteins in the LLPS field. Their observations suggest that not all proteins that phase separate are governed by the same the driving forces. Moreover, authors validate their findings seen by fitting the predictions against experimental phase diagrams for a set of sequences of hnRNA1, and by using a more realistic and accurate ABSINTH model.

Authors present a novel and efficient way to evolve naturally occurring phase-separating protein sequences, however my only concern regards to generalization of this method to other proteins and its robustness. Since only few protein sequences have been used in this work, I suggest extending the discussion about how this method could be applied at large-scale on the current LLPS databases (e.g., PhasePro, PhaSepDB, etc). This may allow to identify group of proteins that exhibit LLPS and are governed by similar driving forces. After addressing this minor comment, I recommend this work to be published in PLOS Computational Biology.

**Have the authors made all data and (if applicable) computational code underlying the findings in their manuscript fully available?**

Reviewer #1: None

Reviewer #2: **No: **The authors mention that the data will be made available upon acceptance. Therefore, I cannot comment on whether all required material will be provided.

Reviewer #3: Yes

PLOS authors have the option to publish the peer review history of their article (what does this mean?). If published, this will include your full peer review and any attached files.

Reviewer #1: No

Reviewer #2: **Yes: **Vasilis J Promponas

Reviewer #3: No

Figure Files:

Data Requirements:

Reproducibility:

References:

---

## [Editor Report · Decision Letter 1]

7 Aug 2021

Dear Dr Reinhardt,

We are pleased to inform you that your manuscript 'Targeted modulation of protein liquid-liquid phase separation by evolution of amino-acid sequence' has been provisionally accepted for publication in PLOS Computational Biology.

Best regards,

Damiano Piovesan

Guest Editor

PLOS Computational Biology

Arne Elofsson

Deputy Editor

PLOS Computational Biology

---

## [Editor Report · Acceptance letter]

16 Aug 2021

PCOMPBIOL-D-21-00779R1 

Targeted modulation of protein liquid–liquid phase separation by evolution of amino-acid sequence

Dear Dr Reinhardt,

I am pleased to inform you that your manuscript has been formally accepted for publication in PLOS Computational Biology. Your manuscript is now with our production department and you will be notified of the publication date in due course.

With kind regards,

Zsofi Zombor
